# An Investigation of Factors Influencing Tool Life in the Metal Cutting Turning Process by Dimensional Analysis [†]

**Sara M. Bazaz** [1,*] , **Juho Ratava** [1] , **Mika Lohtander** [2] and **Juha Varis** [1]

1  Department of Mechanical Engineering, Schools of Energy Systems, LUT University, 53850 Lappeenranta, Finland
2  School of Technology Industry, Turku University of Applied Science, 20520 Turku, Finland
*  Correspondence: sara.bazaz@lut.fi
†  This paper is an extended version of our paper published in Flexible Automation and Intelligent Manufacturing: The Human-Data-Technology Nexus. In Proceedings of the FAIM 2022, Detroit, MI, USA, 19–23 June 2022.

**Abstract:** This article uses dimensional analysis to formulate the tool life in the turning process of metal cutting for small-lot production by considering the impacts of the most important parameters. The estimation of tool life specifies process efficiency, machining productivity, resource consumption, machining time, and cost. Many parameters influence tool life on the real shop floor in small-lot production. This literature review studies 29 parameters affecting tool life directly or indirectly. The results of this research are represented as a graph-based analysis in the form of a web of interdependencies and a relationship matrix. The relationship matrix illustrates the direct and indirect interdependencies of the parameters which influence tool life in the turning process. The graph visualizes the weight of the parameters for the estimation of tool life in small-lot production. A cause-and-effect diagram is extracted from the relationship matrix to study the parameters affecting tool life in small-lot production. A dimensional analysis is executed based on the cause-and-effect diagram in order to calculate the tool life. The functions of tool life involve the cutting conditions, tool and workpiece hardness, cutting force, and cutting temperature. The dimensional analysis shows that the cutting speed, feed rate, and workpiece hardness are the most effective factors impacting tool life in the turning process.

**Keywords:** tool life; small-lot production; turning process; dimensional analysis



## 1. Introduction

Machining is the process of removing material from the workpiece to form the desired part. Metal cutting is one of the typical manufacturing processes used in industry. There are several methods that can be used to cut the material. The turning process is one of the most widely used machining processes in the manufacturing industry. Basically, in the turning process, the stationary tool, through feed movement, removes the rotating material workpiece. During the cutting process, tool wear occurs due to friction between the tool and the workpiece, high stress on the tool nose, and chip sliding along the rake face. A gradual increase in tool wear can cause tool failure. Tool failure can cause crucial problems and the waste of resources such as time and costs. The measure of the usable cutting time of the tool or, in other words, the remaining tool life is essential for optimizing the tool life in production and preventing tool breakage during the machining process [1,2].

The performance and tool life of cutting tools are highly influenced by the tool materials and coatings used to manufacture the tool insert [3]. There are various studies that aimed to simulate, detect, and predict the tool wear state and optimal parameters for a given tool–material combination [4–6]. Some studies have attempted to predict the total tool life for a given material [7], whereas other studies aimed to predict the remaining tool life for a given partially used tool [6]. Mathematical modeling, the Taylor equation,

regression analysis, machine learning, image recognition, and deep learning are all methods used for this purpose [8–10].

A graph-based analysis of parameters affecting tool life in small-lot production was presented by the authors of [11]. This paper expands on the authors' initial paper by providing a more detailed overview of the parameters affecting tool life in the turning process of metal-cutting and extracting a cause–effect diagram from the results. According to the cause–effect diagram, the dimensional analysis demonstrates the functional relationship between the tool life and cutting parameters, workpiece and tool hardness, cutting force, and cutting temperature.

In manufacturing industries, the machining process is used to manufacture products for different lot sizes. The lot size is defined as the quantity of a product to be produced in a single production run [12]. According to the quantity of the product in the lot, it is classified into three groups: large-size, medium-size, and small-size. In the case of a large lot, a product is produced continuously over a long period of time. Large-lot production can follow the regular rules and experimental setups and standards [13]. It requires the same setup and equipment in the production line. All the final products have the same quality level. It consumes less energy compared to the other lot sizes. It is easier to predict and reduce the impact of production on the environment by optimizing the production parameters. Small-lot production is the manufacturing of products in medium-sized lots [14]. The size of the products varies according to the order. The production process is defined for a particular product and is not repetitive. In small-lot production, products can be customized. It does not require a large production line and can be performed by small businesses. Although small-lot production can be cheaper for a company, the percentage of costs such as the production costs, labor costs, and energy costs required for producing one product is higher than that in large-lot production. Small-lot production requires expertise in calculating the production factors depending on environmental changes in the manufacturing shop [15].

Extensive research has been conducted on the Nordic manufacturing industry. The Nordic manufacturing sectors have four characteristics. Firstly, in the Nordic countries, manufacturing sectors are widely distributed across the country to maintain a balance in job creation between large cities and small towns. Secondly, many manufacturing companies are small, but multinational companies have added value for research and development, especially in Finland and Sweden [16]. Business-to-business suppliers play an important role in the Nordic manufacturing sectors. In recent decades, there has been a surge in the number of highly skilled workers [17]. In 2004, one project investigated the situation of metal-based manufacturing for small-lot production in Nordic countries. The project aimed to propose cost-effective methods for the manufacturing of various products by small- and medium-sized manufacturing companies [18].

An entire tool life can be between 15 and 40 min. In small-lot production, a tool may be used for just few seconds or minutes of its tool life, since production planning is mostly under the influence of human experience and knowledge, and optimizing the cutting conditions changes the energy consumption by 6% to 40% [19]. In the manufacturing industry, 20% of downtime is due to tool failure, which affects 30% of the machining costs [20,21] The monitoring and estimation of accurate tool wear facilitate small- and medium-sized enterprises in increasing their productivity, saving resources, and reducing the downtime and final cost of the machining. Many sectors, such as spare part and tailored product production and the maintenance of machine equipment, could benefit from the calculation of tool life while constantly changing the workpiece materials and conditions.

Efficiency, energy consumption, cost, and time usage are defined by the tool life in small-lot production [22]. Many factors, including the cutting condition parameters, cutting force, and cutting temperature, have direct impacts on tool wear and, as a result, tool life [23]. These characteristics, such as the cutting speed and depth of cut, have impacts on other variables, such as chip formation. They are selected based on the geometry and material of the tool, as well as the geometry and material of the workpiece [24]. In order

to conserve resources, for example, by preventing unnecessary tool changes, small-lot production (less than 10 products) can be utilized to determine all the key aspects affecting tool life.

Small-lot production involves more factors than large-lot production. There may be variations in the choice of cutting tools and cutting circumstances depending on the quantity of the given product and the manufacturing of various products [25]. For example, if a company needs to produce ten identical products, it can choose a suitable tool holder, insert, and cutting parameters to optimize energy consumption and machine depreciation. However, if the company needs to produce several different products, it can select a suitable tool for all of them. This tool may be the same or different from the one used for the previous batch, containing similar or some of the same products [26].

This research aims to demonstrate the complications involved in manufacturing data and identify a mathematical model so as to estimate tool life in the manufacturing of small batch sizes by considering factors other than the cutting conditions. This research concentrates on data related to tool life in the turning process of metal materials. The method of this research is a literature review of books, conference papers, and journal papers. The results of the study are shown in the form of a relationship matrix. The matrix contains 29 factors with direct and indirect effects on the tool life. The weights of the factors are shown in a graph-based analysis. The weight represents the importance of the parameters affecting tool life. The parameters driving the tool life formula were chosen based on the weights in the graph-based analysis.

The contribution of this research is a study of tool life using different workpieces with the same tool to produce a limited number of various parts. During machining, it is crucial to prevent tool breakage or to use a worn tool for manufacturing. In machining, different workpieces change other parameters, such as the cutting force, friction, vibration, chip formation, chip thickness, and tool wear. In this research, we studied tool life in variable environmental conditions. Many researchers have studied tool life while machining a specific workpiece in a stable environment [8]. It is important to predict tool wear behavior while machining different workpieces under various cutting conditions. This paper investigates a tool life formula for small-lot production. In contrast to other studies which use the cutting speed, feed rate, and depth of cut for one workpiece material [5,9,23], in this research, the workpiece and tool properties are directly factored into the tool life function. The dimensional analysis represents the fact that, when calculating the tool life, the workpiece hardness influences the feed rate, cutting force, and cutting temperature.

The rest of this paper is organized as follows. The literature review protocol and reference analysis are described in Section 2. Section 3 describes the parameters affecting tool life in the turning process of metal cutting and summarizes them through the relationship matrix, graph-based analysis, and cause–effect diagram. The dimensional analysis uses the parameters from the graph-based analysis and the cause–effect diagram to calculate the tool life. Section 4 discusses the results of the research and their applications and advantages for small-lot production. Section 5 concludes the results of the study.

## 2. Methodology

### 2.1. The Scope of the Analysis and Review Methodology

This paper explores the parameters that affect tool life in the turning process of metal materials. The analysis is conducted based on a literature review, and connections between various factors affecting the production are incorporated into a relationship matrix defining an interconnected bidirectional graph. The clustering, impact analysis, and visualization were performed using the Gephi application [27].

The literature review was conducted based on the review protocol introduced by Kitchenham et al. [28]. The protocol involves six steps, forming the framework of this literature review. These steps are research questions, the search process, inclusion and exclusion criteria, quality assessment, data collection, and data analysis.

The research question step concerns the research questions that are addressed in the literature review. The basics of metal cutting and tool life are explained in books and handbooks. However, many parameters with complex relationships are explored to study tool life in the context of metal cutting, but there is no chart available to classify the parameters affecting tool life. In this research, we study the parameters affecting tool life in the metal cutting turning process and represent comprehensive results in the form of a matrix to help researchers to use them in their studies for different purposes. The research question and research problem are as follows:

RQ1: Which factors are needed to estimate an accurate tool life for small-lot production?

According to the research problem and the lack of studies related to small-lot production, the research problem is:

RP1: What are the parameters affecting tool life needed to develop a tool life equation by dimensional analysis?

The formulation of this research question and problem aims to precisely define the framework of this paper.

The search process included the method used to collect the database. An adequate database search was conducted to answer the research question. Scopus provides a large number of scientific articles from a variety of publishers. The literature review focused on studies from 2000 to 2022. The keywords used to conduct the search covered the main parameters that influence tool life. Different key word combinations were searched in Scopus. The selection of keywords used in the construction of this research referred to the group of parameters related to tool life in the turning process. The keywords were parameters affecting the tool life, cutting parameters, tool geometry, cutting force, cutting temperature, chip formation, and flank wear. Books were the support materials used to demonstrate the relationships between these parameters, such as the relationship between the tool geometry and the cutting force or the link between the workpiece material and cutting conditions. In total, 275 articles were obtained. Duplicates were removed based on their DOIs.

The inclusion and exclusion criteria established the limitations required to keep the articles relevant so as to answer the research question. The inclusion criteria (IC) were chosen to cover the scope of the research, and the exclusion criteria (EC) covered the out-of-scope areas. These criteria are listed below:

IC1: The paper is about the turning operation of metal material.

IC2: The paper studies parameters affecting tool life and tool wear.

IC3: The paper contains mathematical modelling or a formula used to calculate the tool life or related parameters.

IC4: The paper is written in English.

Exclusion criteria:

EC1: The paper is not about methods of monitoring tool wear or tool life.

EC2: The paper is not about AI methods used to estimate tool life. These studies investigated methods for simulating tool life or tool wear. They are helpful for determining certain links between a few parameters, but they do not explain the nature of the relationship.

EC3: The paper is not about milling or other processes.

In the third step, the inclusion and exclusion criteria were applied to the titles and abstracts of the articles. Then, all the papers were checked manually to uphold the inclusion criteria. The article quality assessment involved more quality indicators so as to keep the papers that answered the research question. In the quality assessment, we ensured that the paper studied the research question in general terms. Figure 1 demonstrates the data collected based on the four steps. The papers which only studied surface roughness are excluded from this study. Articles containing words such as surface roughness or morphology, together with other parameters affecting tool life, were included in the research. In the last step, data analysis was performed on 101 articles to demonstrate the studies' progress, countries interested in this topic, and the parameters which are most widely studied.

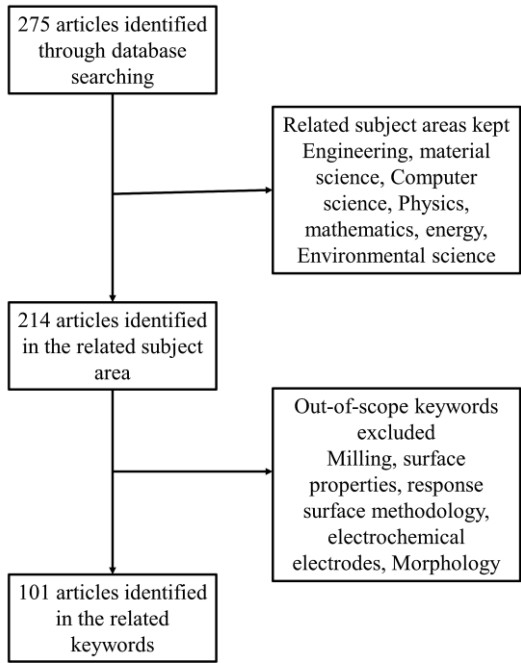

**Figure 1.** Methodology applied to identify 101 articles.

After selecting the appropriate papers from the last three steps among the data collection and data analysis steps, we performed a descriptive analysis and an in-depth analysis. Bibliometric analysis is a subset of descriptive analysis. The in-depth analysis included the parameters involved in tool life calculation, the relationship matrix, and the graph-based analysis, which will be discussed in the Results section.

### 2.2. Bibliometrics Analysis

As Figure 1 illustrates, in total, 101 papers were identified from the research database and the inclusion and exclusion criteria stages. The purpose of this analysis is to describe the importance of studying tool life, the parameters affecting tool wear or tool life, and the pattern of this area of study over the last 22 years.

Figures 2 and 3 display the distribution of the number of publications per year. The plot shows that more attention has been devoted to the topic in the last decade, especially after 2016. However, 63% of articles are from 2015 to 2022. This result shows that although the topic is considered as fundamental for metal cutting, it requires attention in regard to the production of new materials, environmental changes, and new technologies. The sudden rise in the number of publications after 2017 represents the essence of the subject in recent years.

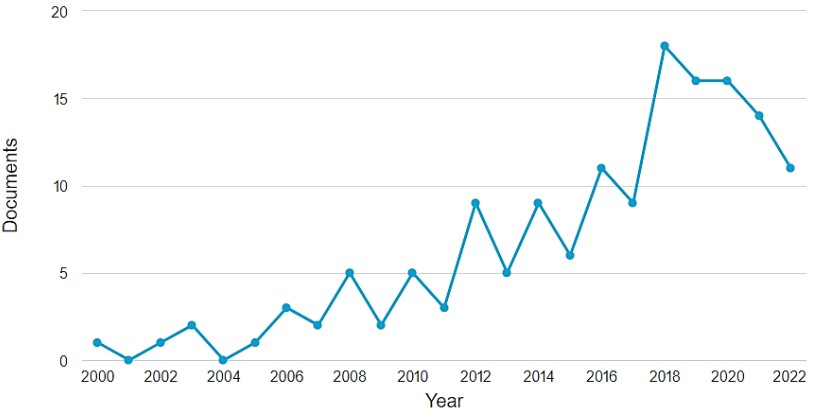

**Figure 2.** Number of publications per year according to Scopus search results.

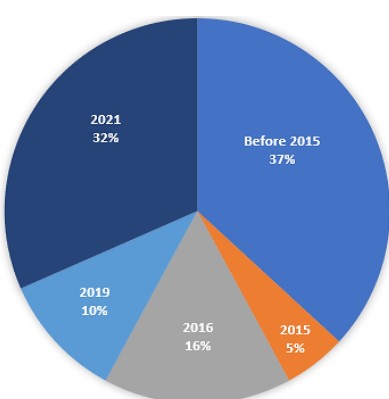

**Figure 3.** Distribution of the publications from 2015 to 2021 according to Scopus search results.

Figure 4 illustrates the types of reviewed scientific documents. Almost 90% of the documents are scientific articles or conference papers. The studied books account for 2% and review papers account for 5% of the reviewed documents. Figure 5 shows the areas of these studies. Most areas refer to engineering, materials science, physics, and computer science.

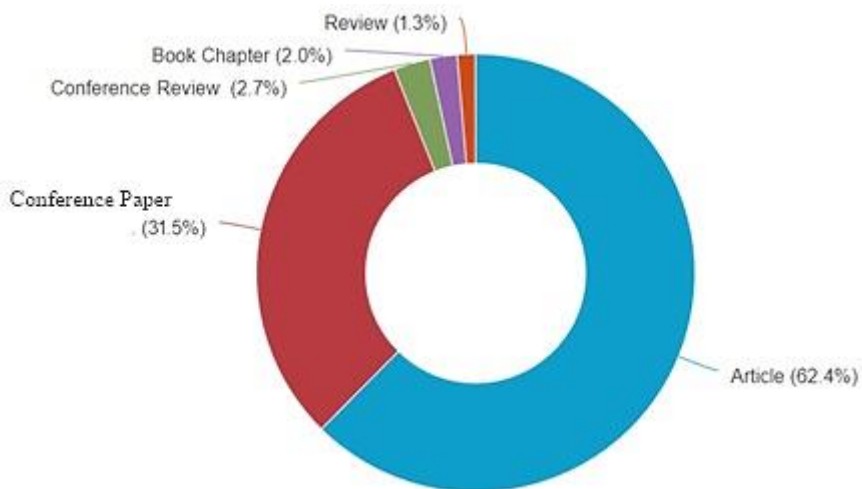

**Figure 4.** Document types of the publications according to Scopus search results.

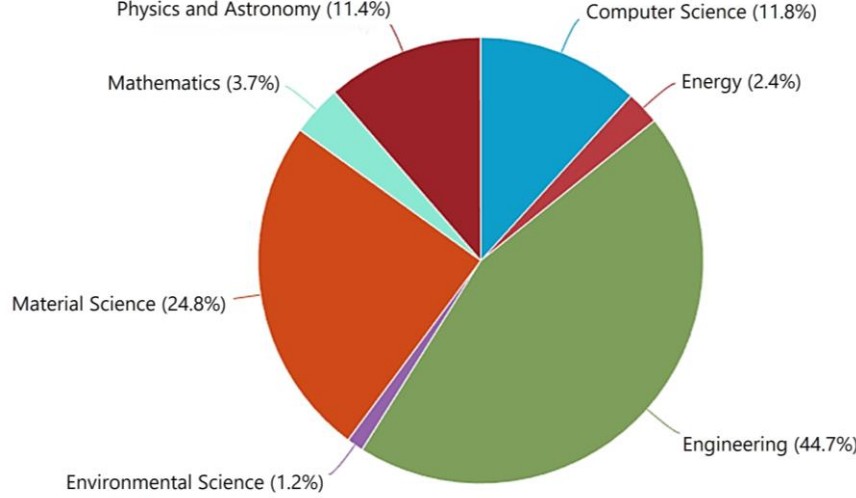

**Figure 5.** Fields of the publications according to Scopus search results.

Figure 6 shows the top ten countries that have performed research regarding parameters affecting tool life. This information indicates the need of the industries in these countries to estimate the accurate tool life in manufacturing. Figure 7 is extracted from the database subjects and contexts based on the manual consideration of key words. The bar chart shows which parameters are studied most frequently and have the most significant effects on the tool life, as well as parameters which need further study in the future. As expected, the cutting speed is the most important and most frequently studied parameter. On the other hand, the workpiece material and vibration are the least frequently studied factors.

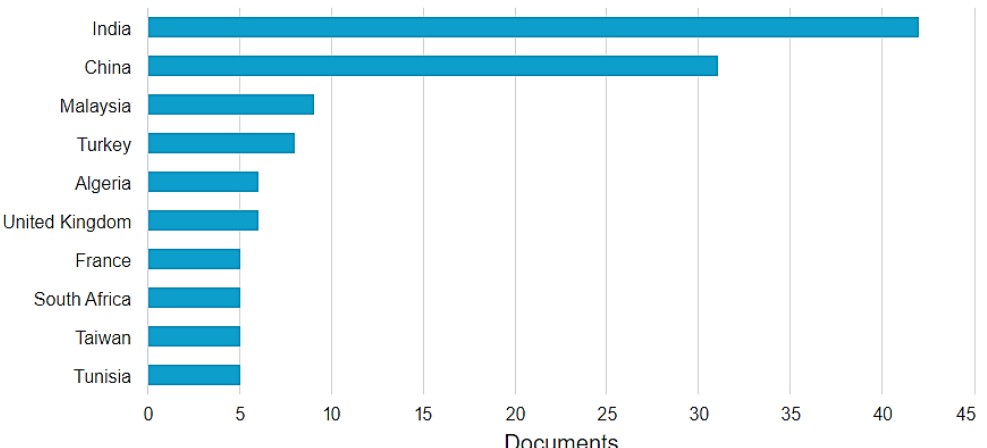

**Figure 6.** Top 10 countries among the publications according to Scopus search results.

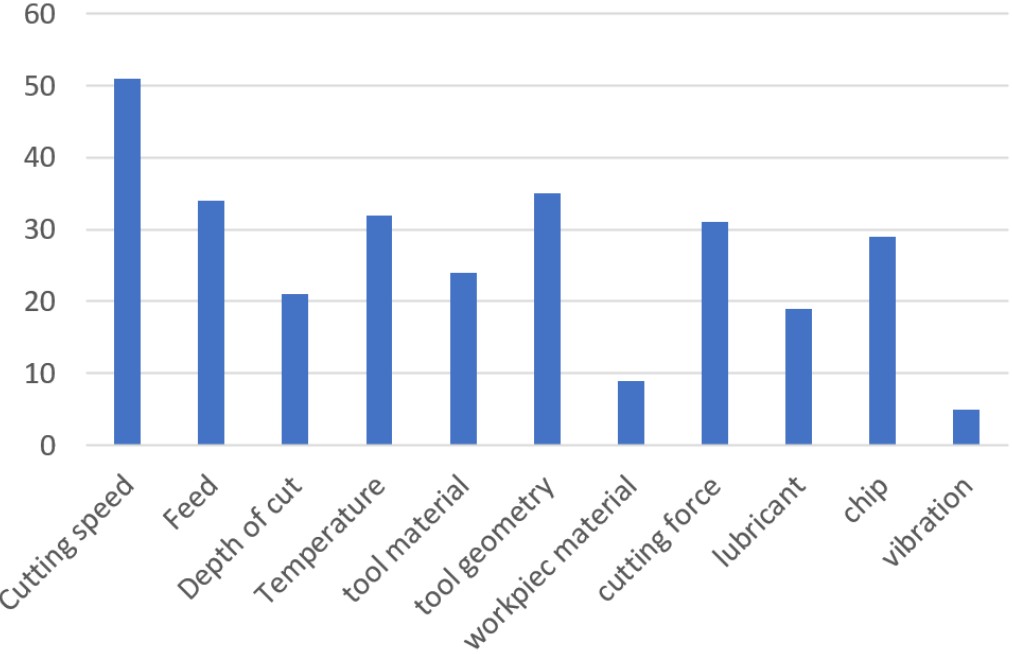

**Figure 7.** Number of references that studied important parameters affecting tool life in the database based on keywords (one paper may be in more than one category).

### 2.3. Dimensional Analysis

One of the applications of this study is the use of dimensional analysis to derive a tool life formula considering the effects of workpiece changes. Dimensional analysis a technique for identifying correlations between physical quantities in a problem by comparing their dimensions, without having to solve the problem in its entirety. It is useful when two quantities are directly proportional to one another and one of the quantities needs to be converted into the other using a common equivalent, conversion factor, or conversion

relation. The given quantity is the problem's starting point, the desired quantity is the answer to the problem, the unit path is the series of conversions required to obtain the desired quantity, and the conversion factors are the equivalents required for conversion between different measuring systems and the removal of any unnecessary units from the problem. There are five steps used to execute dimensional analysis:

1. Determine the given quantity for the problem.
2. Determine the answer to the problem (desired quantity).
3. Using equivalents as conversion factors, establish the unit path from the given quantity to the desired quantity.
4. Configure the conversion factors so that the cancelled units can be returned.
5. To determine the numerical value of the desired quantity, multiply the numerators, the denominators, and the product of the numerators by the product of the denominators.

## 3. Result

### 3.1. Research-Literature Review

This section presents the general parameters that affect tool life in the metal cutting turning process. The aim of this study is to show that there are relationships between the parameters rather than how these relationships work in different special cases. The relationships between the parameters are shown in the form of a matrix and complex web. Figure 8 illustrates the connections between the parameters affecting tool life. The size of the nodes shows the weight of the factors affecting the tool life.

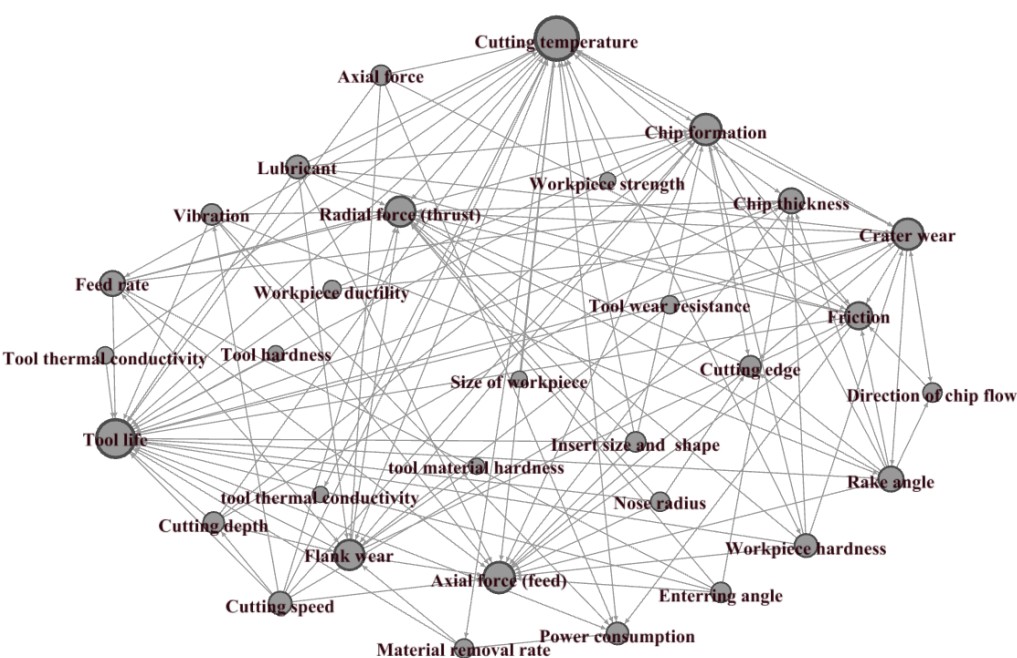

**Figure 8.** Graph-based analysis of parameters affecting tool life in the turning process.

Table 1 demonstrates the relationship matrix of the studied parameters. The matrix dimensions are 29 × 23, and the parameters in the rows affect parameters in the columns. In the matrix, number 1 shows that there is a relationship, and 0 indicates that there is no relationship. The purpose of forming the matrix is to extract the relationship groups in order to study and estimate the tool life in small-lot production using different approaches.

**Table 1.** Relationship matrix of the parameters affecting tool life in the turning process.

| | Workpiece Hardness | Tool Thermal Conductivity | Tool Wear resistance | Tool Material Hardness | Nose Radial | Rake Angle | Cutting Edge | Flank Wear | Crater Wear | Feed Rate | Cutting Depth | Material Removal Rate | Axial Force (Feed) | Radial Force (Thrust) | Cutting Temperature | Chip Formation | Chip Thickness | Direction of Chip Flow | Lubrication | Power Consumption | Vibration | Friction | Tool Life |
|---|---|---|---|---|---|---|---|---|---|---|---|---|---|---|---|---|---|---|---|---|---|---|---|
| Workpiece hardness | 0 | 0 | 0 | 0 | 0 | 0 | 0 | 0 | 1 | 0 | 0 | 0 | 1 | 1 | 1 | 1 | 1 | 0 | 0 | 0 | 0 | 0 | 1 |
| Workpiece strength | 0 | 0 | 0 | 0 | 0 | 0 | 0 | 0 | 0 | 0 | 0 | 0 | 0 | 0 | 1 | 0 | 0 | 0 | 0 | 0 | 0 | 0 | 0 |
| Workpiece ductility | 0 | 0 | 0 | 0 | 0 | 0 | 0 | 0 | 1 | 0 | 0 | 0 | 0 | 0 | 0 | 1 | 0 | 0 | 0 | 0 | 0 | 0 | 1 |
| Tool thermal conductivity | 0 | 0 | 0 | 0 | 0 | 0 | 0 | 0 | 0 | 0 | 0 | 0 | 0 | 0 | 1 | 0 | 0 | 0 | 0 | 0 | 0 | 0 | 1 |
| Tool wear resistance | 0 | 0 | 0 | 0 | 0 | 0 | 0 | 1 | 1 | 0 | 0 | 0 | 0 | 0 | 0 | 0 | 0 | 0 | 0 | 0 | 0 | 0 | 1 |
| Size of workpiece | 0 | 0 | 0 | 0 | 0 | 0 | 0 | 0 | 0 | 0 | 0 | 0 | 0 | 0 | 1 | 0 | 0 | 0 | 0 | 0 | 0 | 0 | 0 |
| Insert size & shape | 0 | 0 | 0 | 0 | 0 | 0 | 0 | 0 | 0 | 0 | 0 | 0 | 1 | 1 | 1 | 1 | 0 | 0 | 0 | 0 | 0 | 0 | 1 |
| Tool hardness | 0 | 0 | 0 | 0 | 0 | 0 | 0 | 1 | 0 | 0 | 0 | 1 | 0 | 0 | 0 | 0 | 0 | 0 | 0 | 0 | 0 | 0 | 1 |
| Nose radius | 0 | 0 | 0 | 0 | 0 | 0 | 1 | 0 | 0 | 0 | 0 | 0 | 1 | 1 | 0 | 0 | 0 | 0 | 0 | 0 | 0 | 0 | 1 |
| Rake angle | 0 | 0 | 0 | 0 | 0 | 0 | 1 | 0 | 0 | 0 | 0 | 0 | 1 | 1 | 0 | 1 | 1 | 1 | 0 | 0 | 0 | 1 | 1 |
| Entering angle | 0 | 0 | 0 | 0 | 0 | 0 | 0 | 0 | 0 | 1 | 0 | 0 | 1 | 1 | 0 | 0 | 1 | 0 | 0 | 0 | 0 | 0 | 1 |
| Cutting angle | 0 | 0 | 0 | 0 | 0 | 0 | 0 | 0 | 0 | 0 | 0 | 0 | 1 | 1 | 0 | 0 | 0 | 0 | 0 | 0 | 0 | 0 | 0 |
| Flank wear | 0 | 0 | 0 | 1 | 0 | 0 | 0 | 0 | 0 | 1 | 0 | 0 | 1 | 1 | 1 | 0 | 0 | 0 | 0 | 0 | 0 | 1 | 1 |
| Crater wear | 0 | 0 | 0 | 0 | 0 | 1 | 1 | 0 | 0 | 0 | 0 | 0 | 1 | 1 | 1 | 0 | 0 | 1 | 0 | 0 | 0 | 1 | 1 |
| Cutting speed | 0 | 0 | 0 | 0 | 0 | 0 | 0 | 1 | 1 | 0 | 0 | 1 | 1 | 1 | 1 | 1 | 0 | 0 | 0 | 0 | 1 | 0 | 1 |
| Feed rate | 0 | 0 | 0 | 0 | 0 | 0 | 0 | 1 | 0 | 0 | 0 | 0 | 1 | 1 | 1 | 1 | 1 | 0 | 0 | 0 | 0 | 0 | 1 |
| Cutting depth | 0 | 0 | 0 | 0 | 0 | 0 | 0 | 1 | 0 | 0 | 0 | 0 | 0 | 0 | 1 | 1 | 1 | 0 | 0 | 0 | 0 | 0 | 1 |
| Material removal rate | 0 | 0 | 0 | 0 | 0 | 0 | 0 | 1 | 0 | 0 | 0 | 0 | 0 | 0 | 0 | 0 | 0 | 0 | 1 | 0 | 0 | 0 | 1 |
| Axial force | 0 | 0 | 0 | 0 | 0 | 0 | 0 | 1 | 0 | 0 | 0 | 0 | 0 | 0 | 1 | 0 | 0 | 0 | 0 | 1 | 0 | 1 | 1 |
| Radial force | 0 | 0 | 0 | 0 | 0 | 0 | 0 | 1 | 0 | 0 | 0 | 0 | 0 | 0 | 1 | 0 | 0 | 0 | 0 | 1 | 0 | 1 | 1 |
| Cutting temperature | 1 | 1 | 1 | 0 | 0 | 0 | 0 | 1 | 1 | 1 | 0 | 1 | 1 | 1 | 0 | 1 | 1 | 0 | 0 | 1 | 0 | 0 | 1 |
| Chip formation | 0 | 0 | 0 | 0 | 0 | 0 | 1 | 1 | 1 | 0 | 0 | 0 | 1 | 0 | 1 | 0 | 0 | 0 | 0 | 0 | 0 | 1 | 1 |
| Chip thickness | 0 | 0 | 0 | 0 | 0 | 0 | 0 | 0 | 0 | 0 | 0 | 0 | 1 | 1 | 1 | 0 | 0 | 0 | 0 | 0 | 0 | 0 | 1 |
| Direction of chip flow | 0 | 0 | 0 | 0 | 0 | 0 | 0 | 0 | 1 | 0 | 0 | 0 | 0 | 0 | 0 | 0 | 0 | 0 | 0 | 0 | 0 | 1 | 0 |
| Lubrication | 0 | 0 | 0 | 0 | 0 | 0 | 0 | 1 | 1 | 0 | 0 | 0 | 1 | 1 | 1 | 1 | 0 | 0 | 0 | 0 | 0 | 1 | 1 |
| Power consumption | 0 | 0 | 0 | 0 | 0 | 0 | 0 | 0 | 0 | 0 | 0 | 0 | 0 | 0 | 1 | 0 | 0 | 0 | 0 | 0 | 0 | 0 | 0 |
| Vibration | 0 | 0 | 0 | 0 | 0 | 1 | 0 | 0 | 0 | 0 | 0 | 0 | 1 | 1 | 0 | 0 | 0 | 0 | 0 | 0 | 1 | 0 | 1 |
| Friction | 0 | 0 | 0 | 0 | 0 | 0 | 0 | 0 | 0 | 0 | 0 | 0 | 1 | 1 | 1 | 0 | 0 | 0 | 0 | 1 | 0 | 0 | 1 |
| Tool life | 0 | 0 | 0 | 0 | 0 | 0 | 0 | 0 | 0 | 0 | 0 | 0 | 0 | 0 | 0 | 0 | 0 | 0 | 0 | 1 | 0 | 0 | 0 |

Figure 9 shows an example of the extracted groups of relationships to form a cause-and-effect diagram. The values on the left side of the factoring line indicate that these parameters affect the factoring line, while the values on the right side of the line indicate that the subject factor affects those values. In the following analysis, these groups of parameter relationships are expressed.

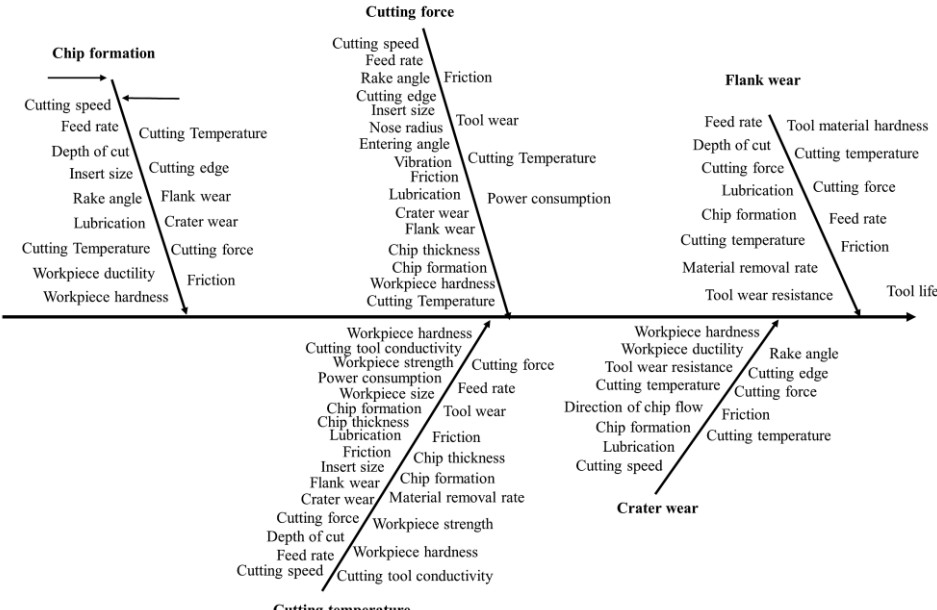

**Figure 9.** The cause-and-effect diagram of parameters affecting tool life in the turning process of metal cutting.

The tool life has different definitions based on the scale of time in which the tool operates or the usable time before the tool reaches the criterion value of flank wear [29]. Many researchers have studied the effects of cutting conditions on the tool life, as these parameters are known to be useful for calculating tool life [30,31]. The Taylor equation (Equation (1)) is one of the most popular equations based on cutting conditions. Equation (1) is the extended Taylor equation, where $v_c$ is the cutting speed, $f$ is the feed rate, $d$ is the depth of cut, and $l$, $m$, $n$, and $\varepsilon$ are constants [32].

$$T = C\left(v_c{}^l f^m d^n\right)\varepsilon \tag{1}$$

This paper shows that the cutting speed affects the cutting temperature, cutting force, chip formation, vibration, and tool wear. This means that this parameter has a complicated influence on the tool life [33]. For this paper, we selected five groups of related parameters that have direct effects on the tool life. In the following analysis, these groups are expressed to show why these parameters were selected.

Tool wear occurs as the gradual loss of material in a tool during a cutting operation that causes changes in the shape of the tool [34]. Contact between the cutting tool and the workpiece and contact and sliding between the cutting tool and chip are the reasons for the production of tool wear [35]. These types of contact cause stress, friction, deformation, and high temperatures acting on the cutting tool [36]. To predict tool wear, it is important to know the tool wear mechanism. Any factor changing the contact conditions can influence tool wear [37]. During machining, tool wear occurs due to abrasion, diffusion, adhesion, oxidation, and fatigue [38].

Mechanical abrasion occurs between the workpiece material and the cutting tool when hard particles on the underside of the chip pass over the tool, which is called abrasion wear [39,40]. This wear is relatively predictable and results in a stable tool life [41]. It is predominant in cases where prevailing strain results in hard spots throughout the point contact zone of the interface. Formula (2) describes the abrasive wear volume, where $k_\alpha$ is the abrasion constant, $S$ is the distance of sliding, $N$ is the normal load, $K$ is the probability of forming a sizable wear particle, $H_a$ is the hardness of the abrasive particles of the work

material, $H_t$ is the hardness of the tool, and $n$ is the ratio of the tool hardness to the abrasive particles defining the workpiece hardness [42].

$$V_{ab} = \frac{k_\alpha \times S \times N}{K} \times \frac{H_a^{n-1}}{H_t^n}$$ (2)

Adhesion wear is formed on the interface of the chip and tool material due to the friction mechanism [43]. When the intersection is fractured, small particles of the tool material are torn away and move to under the chip or to the new workpiece surface [44]. Usually, adhesion occurs at high pressures and temperatures at the tool cutting edge, because these conditions form a metallic bond-like spot weld at the tool–chip interface. The spot weld forms built-up edges due to the irregular chip flows. The chip slides over the built-up edges and then causes the fracture of the tool cutting edge [45].

Diffusion wear is the result of the diffusion process, where atoms in a metallic crystal lattice move from the region of high atomic concentration to the region of low concentration [46]. During metal cutting, the temperature between the tool and workpiece rises. The high temperature causes movement of the atoms from the tool material to the workpiece materials, thereby weakening the structure of the tool [47]. Chemical conjunction between the workpiece material and the cutting tool material causes chemical interactions. Oxidation is the result of a chemical reaction between the tool face and oxygen [48].

Tool wear influences the power applied for cutting, the finishing quality, the cost of machining, and the tool life of the cutting tool [49]. Flank wear occurs when the relief face of the tool rubs against the workpiece surface. Flank wear influences the finish and dimensional accuracy of the final product [50]. Flank wear is the most common and dominant types of wear which occurs on any tool materials and in any machining context, as it is caused by adhesion and abrasion between the tool flank and the workpiece surface [51]. High temperature between the tool and workpiece material increases adhesion [52].

Formula (3) represents the relationship of flank wear with abrasive and adhesive wear. In Formula (1), w is the flank wear land width, $H$ is the cutting tool material hardness, $v_s$ is the sliding speed, $\sigma_n$ is the normal stress between the workpiece and the tool flank face, and $A$ is the abrasive/adhesive wear constant [29].

$$\frac{dVB}{dt} = \frac{A}{H}\sigma_n v_s$$ (3)

In some conditions, it is necessary to account for diffusion when calculating flank wear. Formula (2) demonstrates the relationship between flank wear and diffusion wear, where E is the process activation energy, $R$ is the universal gas constant, $T_f$ is the cutting temperature in the tool flank zone, and $B$ is the diffusive wear constant [29].

$$\frac{dVB}{dt} = B exp\left(\frac{-E}{RT_f}\right)$$ (4)

The complete form of the equation used to calculate the flank wear when considering abrasive, adhesive, and diffusive wear is expressed in Formula (5), where $v_s$ is the sliding speed, and $F_f$ is the feed rate [20].

$$\frac{dVB}{dt} = \frac{A}{H}\sigma_n v_s + B exp\left(\frac{-E}{RT_f}\right)$$ (5)

Usually, researchers measure flank wear as an indicator of tool life [7]. Increasing the cutting velocity or depth of cut, cutting force, and cutting temperature increases flank wear [53,54]. As the cutting speed increases, the cutting temperature increases, which causes larger chip loads on the tool and then causes drastic wear and decreases the tool life [55]. An excessive thermal load causes strong adhesive between the workpiece and tool,

which creates a built-up edge that increases flank wear on the tool [56]. At a low cutting speed, when the cutting temperature is lower than the optimal cutting temperature, the feed increases, and the tool wear rate increases. At an average cutting speed, the cutting temperature passes the optimal cutting temperature, causing an increase in the tool wear rate. At a high cutting speed, the cutting temperature is higher than the optimal cutting temperature, and the tool wear is increased [57,58].

Increasing the flank wear boosts the cutting force [59]. A higher feed induces a greater cutting force on the area of chip–tool contact on the rake face and tool–work contact on the flank face, which increases the cutting temperature [60]. The effect of the depth of cut on tool wear is not significant. However, in a process with a variable depth of cut, the tool wear increases with the increasing cutting depth. Generally, the depth of cut influences the flank wear and chatter [61,62].

During metal cutting, when chips erode the rake face of the tool, crater wear occurs on the rake face of the tool [63]. Chips flow across the rake face and cause severe friction between the chip and the rake face. This does not degrade the use of the tools until it creates cutting-edge failure. Crater wear increases the rake angle and reduces the cutting force; however, it weakens the strength of the cutting edge [64]. Crater wear is mainly created due to diffusion and abrasion [65].

Crater wear is more common in ductile materials because of the production of continuous chips over a long time. It occurs more in tools with high hot hardness, such as high-speed steel tools. The crater depth is the most frequently used parameter for evaluating rake face wear [5]. Crater wear is predominant in high-speed cutting. Machining with carbide tools at a high cutting speed causes adhesion, diffusion, and post-abrasion, which produce crater wear [66].

The properties of the workpiece material play a crucial role in estimating the tool life. The workpiece hardness, for instance, directly affects tool wear due to its abrasive action [67]. It is important to select an appropriate tool geometry, including factors such as the nose radius, based on the workpiece material properties [68]. The tool insert's tip thickness, tip shape, chip formation, and cutting force distribution (feed force and thrust force) are dependent on the hardness of the material [69,70].

Figure 10 illustrates the tool geometry and angles. The rake angle, clearance angle, and nose radius are significant tool geometry characteristics that affect tool life [71]. The geometry of the tool and workpiece determines the rake angle [72]. The tool life increases as the positive rake angle rises until it reaches the optimal rake angle, because there is less contact between the chip and the rake face, which lowers the cutting temperature and cutting force [73]. An excessively large a rake angle (rake angle over the optimal rake angle) weakens the cutting tool edge and decreases the tool life [74].

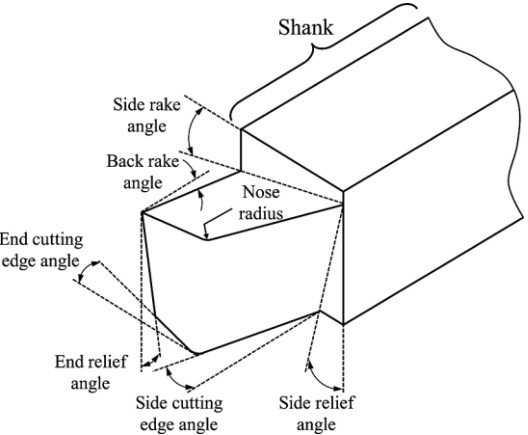

**Figure 10.** Tool geometry and angles- reprinted with permission from [1], Copyright @ 2020, John Wiley & Sons, Inc.

The clearance angle (relief angle) is the angle of orientation of the tool's principal flank surface from the cutting velocity vector and is measured on another plane. The clearance angle avoids rubbing between the tool and the workpiece [58]. Small clearance angles led to large flank wear. If the clearance angle is increased, the temperature and flank wear will decrease, causing an increased tool life. However, an excessively large clearance angle weakens the tool and reduces the tool life, similar to the effect of the rake angle [75].

Based on work material characteristics such as hardness, the nose radius varies [76]. One of the factors affecting chip formation and chip thickness, which affect the tool edge strength, is the nose radius [77]. A large tool nose radius provides a wider area for wear so that there is less wear concentrated on the nose of the insert [78]. The advantage of the larger worn area on the insert is that there is less duplicated wear on the surface of the insert edge [79].

In metal cutting, chips form due to shear deformation of the workpiece along the shear plane [26]. Chip formation is one of the significant factors directly affecting the tool's condition [58]. The chip thickness, chip flow direction, mechanism of the chip, friction between the chips and tool face, and chip temperature determine the tool wear [80]. The plastic deformation of the workpiece, cutting parameters, and tool geometry determine the chip formation type [81]. Cutting conditions, particularly the cutting speed, directly affect workpiece plastic deformation. Increasing the cutting conditions increases the plastic deformation of the workpiece and tool wear until the chip breaks. Increasing chip breakage reduces tool wear [82].

Continuous chipping causes high friction at the tool–chip interface, while a thin chip with a built-up edge (BUE) protects the rake face and reduces tool wear [83]. The cutting speed, depth of cut, rake angle, and cutting fluid are factors that influence the tendency for BUE formation [78]. The chip type affects the dynamic feed force component. In the case of continuous chips, the dynamic feed force is low; in contrast, in the case of discontinuous (broken) chips, the feed force component is larger. Generally, in the case of broken chips, the cutting force is larger than in the case of continuous chips [84].

Cutting forces change during discontinuous chip formation, which can cause chatter and vibration in the machine tool that may cause premature wear of the cutting tool [85]. The direction of the chip flow during the separation of the workpiece surface is dependent on the rake angle [86]. Increasing the curvature of the chip decreases the tool–chip contact and raises the temperature of the tool, especially close to the nose, which increases the tool wear. The workpiece material hardness, cutting tool geometry, and cutting fluids affect chip curl [87].

Formula (6) [88] expresses the chip thickness according to the depth of cut, rake angle, and shear angle. In Formula (2), $t_0$ is the depth of cut, $t_c$ is the chip thickness, $\alpha$ is the rake angle, and $\varphi$ is the shear angle [22]. The shear angle determines the cutting force, the efficiency of the metal removal process, and the surface roughness. By enhancing the contact between the chip and the edge, a thicker chip can occasionally increase the tool life [58]. A good surface quality, low cutting force, and continuous and thin chip production are all correlated with large shear angles [60].

$$r = \frac{t_0}{t_c} = \frac{\sin\varphi}{\cos(\varphi - \alpha)} \tag{6}$$

The entering angle determines the pressure on the cutting edge. The entering angle also impacts the chip thickness and feed rate [30]. An excessively small entering angle produces a thin chip, which reduces the tool life. Formula (7) expresses the relationship between the entering angle and feed rate, in which $\kappa$ is the entering angle, $h$ is the chip thickness, and $f$ is the feed rate [58]. In contrast to a continuous chip, a chip that forms in brittle material is a small segment, with little abrasion on the tool face [89].

$$\sin\kappa = \frac{h}{f} \tag{7}$$

As Figure 11 demonstrates, the cutting force is presented in relation to its three-dimensional components: the feed component in the x-direction, the radial component in the z-direction, and the vertical component in the y-direction [90]. Among these components, the feed and radial components impact flank wear because they are related to the frictional and sliding conditions between the tool and workpiece. The vertical component of the force has a greater effect on the nose wear [91–93].

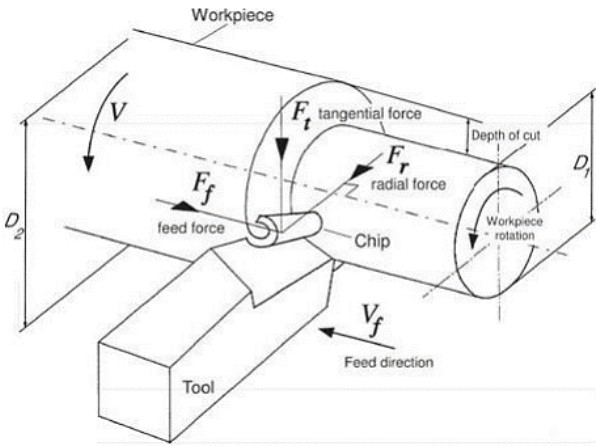

**Figure 11.** Cutting force on the tool nose in the turning process-Copyright © 2001, American Society of Mechanical Engineers [94].

Formula (8) [38] expresses the force component acting on the tool nose and the factors that determine the forces on the tool nose (Figure 11). In Formula (8), $\kappa_1$ is the lubrication constant, $\kappa_m$ is the workpiece material hardness, $\kappa_{t_i}$ relates to the tool geometry associated with the force component, $f$ is the cutting feed (mm/rev), $d$ is the depth of cut (mm), $v_c$ is the cutting speed (m/min), $g_i$ is the wear constant, and $VB$ is the mean wear.

$$F_i = \kappa_i \kappa_m \kappa_t f^a d^b v_c{}^c (1 + g_i VB) \tag{8}$$

The rake angle can control the cutting force. Increasing the back rake angle generates a small shear strain that decreases the cutting force and strength of the tool [85]. In the case of a negative rake angle, the shear strain is higher, but in the practical range of a negative rake angle, the cutting force is higher than that for a positive rake angle. A positive rake angle causes lower deflection acting on the workpiece and tool holder, which reduces the cutting force [95].

Increasing the tool nose radius has direct effects on the thrust and feed components of the cutting force [96]. Edge honing is the sharpening process used to create a controlled radius so as to decrease the friction between the cutting tool and the machined material [97]. Large edge hone tools increase axial and radial forces compared to small edge hone tools [92]. When the tool is worn, the friction between the tool and the workpiece becomes higher, which generates heat, ultimately increasing the cutting forces. An increase in cutting force can cause elastic deformation in the workpiece [98]. Increasing the feed rate boosts the cutting force [99]. The tool life gradually decreases in response to the increasing cutting force and cutting speed (50–125 m/min) [100]. In hard turning, the cutting forces increase with the increase in the chamfer angle [94,95].

Increasing the cutting speed increases cutting force until a certain cutting speed is reached (usually over 150 m/min) [101]. The use of coolant decreases the cutting force due to the increase in hardness of the workpiece, which reduces friction between the tool and chip interface, results in smaller-sized chips, reduces the chip contact length, and prevents the formation of built-up edges [102]. The force ratio is more significant at a low feed rate, and the resultant force is more effective at larger feed rates [62]. The entering angle affects the radial force and axial force, and increasing the entering angle creates a greater axial

force and a lesser radial force. However, decreasing the entering angle may lead to more balanced axial and radial forces [98,99].

During the metal cutting process, the cutting force generates mechanical energy, and part of this energy is converted into heat near the cutting edge of the tool [103,104]. Generally, the heat is generated in the shear zone, rake face, and clearance side of the cutting edge [105,106]. In the machining of ductile materials, the heat resources are caused by shear and plastic deformation, deformation and sliding at tool–chip interfaces, and rubbing at work–tool interfaces [107]. The heat resources raise the temperature at the chip–tool interface. The temperature at the chip–tool interface impacts the cutting force, chip formation mode, and tool life [101,108].

Figure 12 illustrates a force diagram of orthogonal cutting. The cutting force $F_c$ is the equilibrium of the tangential force $F_t$ and feed force $F_f$. The force on the shear plane is the shear force $F_s$. The normal force $F_n$ and friction force $F_u$ are in the secondary shear zone, acting on the tool rake face. In this figure, $\alpha$ is the rake angle, $\varphi$ is the shear angle, and $\beta$ is the friction angle [1].

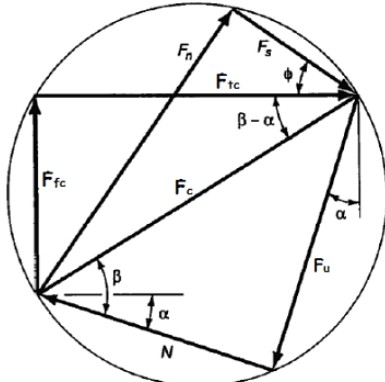

**Figure 12.** Force diagram of orthogonal cutting—Copyright @ 2020, John Wiley & Sons, Inc. [1].

Formula (9) can be used to calculate the mean temperature of the chip from the cutting force ($F_c$), cutting speed ($v_c$), and mechanical equivalent of the heat in the workpiece ($J$), which indicates the amount of mechanical work required to raise a particular mass of water's temperature by one degree.

$$Q = F_c v_c / J \tag{9}$$

Here, $Q$ is the sum of the heat generated on the shear plane and the heat generated by the chip moving over the tool [65].

The generated heat depends on the cutting conditions and workpiece material contact area between the tool and chip, cutting force, and the friction between the tool and workpiece material [109]. A high temperature changes the mechanical properties of the tool and accelerates tool wear [77]. The authors of [110] proposed Formulae (10) and (11) to calculate the average temperature at the tool–chip interface.

$$\Delta T_c = \frac{P_u}{m_c C_s} \tag{10}$$

$$P_u = F_u V_u \tag{11}$$

In this formula, *Pu* is the friction power on the tool face, which is related to the friction force $F_u$ and rake angle $\alpha_r$, $m_c$ is the metal removal rate (kg/s), and $C_s$ is the specific coefficient of the heat of the workpiece (Nm/kg °C). Formula (12) shows the relationship between the friction force, cutting force $F_c$, feed force $F_f$, and rake angle.

$$F_u = F_c \sin\alpha_r - F_f \cos\alpha_r \tag{12}$$

The authors of [111] demonstrated the relationship between the experimental temperature and temperature at the tool–chip interface in Formula (13):

$$\log\left(\frac{\Delta T_m}{\Delta T_c}\right) = 0.06 - 0.195\delta\sqrt{\frac{R_T h_c}{l_t}} + 0.5\log\left(\frac{R_T h_c}{l_t}\right) \tag{13}$$

where $\Delta T_m$ is the maximum temperature rise of the chip at the rake face, which has a contact length of $l_t$. The maximum temperature at the chip–tool interface, substantially, influences the chip formation mode, cutting forces, and tool life [102]. The nondimensional number $\delta$ is the ratio of the plastic layer thickness to the deformed chip thickness $h_c$ at the rake face–chip interface. $R_T$ is a nondimensional thermal number which depends on the cutting velocity, uncut chip thickness, and thermal conductivity of the workpiece.

Low cutting speeds and low rake angles result in a small shear plane angle, consequently increasing the heat flow into the workpiece [112,113]. The heat generated between the workpiece and tool interface has a significant role in restricting the range of the material removal rate when using iron, steel, and high-melting-point alloys as workpieces [114]. The heat generated between the workpiece and tool interface can be measured by the force and chip thickness. The temperature of the chip can be estimated from the heat generation on the shear plane [115,116].

The authors of [77] proposed Formula (14) based on an elliptical-shape heat source with a uniform heat flux distribution between the workpiece and tool flank face. In this formula, $\Delta T_f$ is the temperature rise in the tool flank–workpiece zone, $q$ is the rate of heat supply per unit area, $a_i$ is flank–workpiece contact length, and $k_1$ and $P_{e1}$ are the thermal conductivity and Peclet number of the workpiece material, respectively.

$$\Delta T_f = \frac{2qa_i}{k_1\sqrt{\pi(1.273S_e + P_{e_1}}} \tag{14}$$

The authors of [117] studied tool wear in the machining of Ti–6Al–4V titanium alloy. When the shear strain rate in the shear zone increases, the temperature in the deformation zone increases, which increases the tool wear rate. High temperature in the shear zone causes diffusion, which generates flank and crater wear. The amount of heat conducted in the workpiece is often higher than that calculated using the ideal model, in which heat is generated on the shear plane [118].

A noticeable effect of the heat generated during machining is that on the workpiece material. High temperature reduces the strength and hardness of the workpiece, which decreases the cutting force. Decreasing the cutting force reduces the power consumption [119]. During machining, at a high temperature, adhesion and diffusion wear occur between the workpiece and cutting tool edge. Workpieces with low thermal conductivity cause higher temperatures [120]. The generated heat directly influences the surface roughness. The surface roughness impacts on the wear resistance and fatigue strength [121].

Residual stress is the result of a surface layer's incompatibility with the bulk material. It depends on the workpiece material and cutting parameters. Consequently, residual stresses are produced by any mechanism that modifies the shape or geometry of a surface layer. Theses mechanisms are mechanical (plastic deformation), thermal (thermal plastic flow), and physical (specific volume variation) [122]. The thermal mechanism becomes important as the flank wear rises [123]. The authors of [124] claimed that the load cycle during matching causes residual stress. The feed rate, tool nose radius, and entering angle influence residual stresses. The depth of cut, on the contrary, does not seem to influence residual stresses [125].

In dry machining, the friction at the tool and material interface increases, resulting in high temperature in tool–chip interactions. In addition, higher cutting forces are observed in dry cutting compared to wet cutting. Usually, in dry machining, the tool life is short and the energy consumption is high [126]. The use of lubrication to cool the cutting region and its immediate environment reduces the cutting temperature and friction between the tool

and workpiece [127]. The cutting fluid provides a protective layer on the contact surface of the tool, reducing adhesive wear [128]. Moreover, cooling fluids carry away the chips during the operation, which reduces the friction between the tool–chip interface, thereby improving the machinability and increasing the tool life [118,119].

Minimum quantity lubricant (MQL) reduces flank wear compared to dry machining [129]. The use of MQL reduces the cutting temperature and tool vibration amplitude. Various authors showed that applying 50 mL/h of MQL in high speed machining, with Inconel 718 as a workpiece, improved the tool life by 38 % [130–132]. Studies showed at a high cutting speed (over 100 m/min), the crater wear is higher during wet machining compared to that using MQL [117].

*3.2. Dimensional Analysis*

Dimensional analysis is the mathematical approach to studying the relationships between physical quantities based on their fundamental quantities [133]. Dimensional homogeneity expresses the concept that physical equations have the same dimensions on both sides. The Buckingham theory describes a physical equation with n variables that can be derived equivalently from an equation of n m dimensionless parameters. In this concept, m is the rank of the dimensional matrix. This study considers the SI standard as the base physical dimensions.

$M$ (mass)
$L$ (length)
$T$ (time)

The cause–effect diagram represents the factors that define the tool life in small-lot production. The workpiece material and tool material are the factors defining the cutting parameters, and they influence cutting force, cutting temperature, friction, chip formation, and vibration during the process. As Figure 9 illustrates, material hardness is one of the most effective properties of the material. The factors listed below are considered to take part in the dimensional analysis of the tool life. The dimensions of the tool life and parameters affecting the tool life are as follows:

1. Tool life ($Z$): $T$
2. Tool hardness ($H_T$): $ML^{-1}T^{-2}$
3. Workpiece hardness ($H_w$): $ML^{-1}T^{-2}$
4. Cutting speed ($v_c$): $LT^{-1}$
5. Feed rate ($f$): $L$
6. Depth of cut ($a$): $L$
7. Cutting Force ($F$): $MLT^{-2}$
8. Cutting temperature ($\theta$) : $ML^2T^{-2}$

The dependent parameter is the tool life, and the independent parameters are the tool material hardness, workpiece material hardness, cutting speed, feed rate, depth of cut, cutting force, and cutting temperature. The cutting speed, feed rate, and workpiece hardness are selected as repeating parameters. In this case, n is 8 and m is 3, which means that there are five dimensional groups. Equation (15) shows that the tool life ($Z$) is the function of the selected independent parameters.

$$Z = G(v_c, f, a, H_w, H_T\ F, \theta) \tag{15}$$

Five dimensional groups, $\pi_1, \pi_2, \pi_3, \pi_4, \pi_5$, are derived from Equations (16)–(25).

$$\pi_1 = Z(v_c)^{a_1}(f)^{b_1}(H_w)^{c_1} = T\left(LT^{-1}\right)^{a_1}(L)^{b_1}\left(ML^{-1}T^{-2}\right)^{c_1} \tag{16}$$

The dimension of each quantity should be equal to zero in order to obtain the ultimate exponent of each basic dimension.

$$1 - a_1 - 2c_1 = 0$$

$$a_1 + b_1 - c_1 = 0$$

$$c_1 = 0$$

$$=> a_1 = 1, b_1 = -1$$

$$\pi_1 = Z\frac{v_c}{f} \tag{17}$$

$$\pi_2 = a(v_c)^{a_2}(f)^{b_2}(H_w)^{c_2} = L\left(LT^{-1}\right)^{a_2}(L)^{b_2}\left(ML^{-1}T^{-2}\right)^{c_2} \tag{18}$$

Here,

$$- a_2 - 2c_2 = 0$$

$$- a_2 - 2c_2 = 0$$

$$c_2 = 0$$

$$=> a_2 = 0, b_2 = -1$$

$$\pi_2 = \frac{a}{f} \tag{19}$$

$$\pi_3 = H_T(v_c)^{a_3}(f)^{b_3}(H_w)^{c_3} = ML^{-1}T^{-2}\left(LT^{-1}\right)^{a_3}(L)^{b_3}\left(ML^{-1}T^{-2}\right)^{c_3} \tag{20}$$

$$- 2 - a_3 - 2c_3 = 0$$

$$-1 + a_3 + b_3 - c_3 = 0$$

$$1 + c_3 = 0$$

$$=> a_3 = 0, b_3 = 0, c_3 = -1$$

$$\pi_3 = \frac{H_T}{H_w} \tag{21}$$

$$\pi_4 = F(v_c)^{a_4}(f)^{b_4}(H_w)^{c_4} = MLT^{-2}\left(LT^{-1}\right)^{a_4}(L)^{b_4}\left(ML^{-1}T^{-2}\right)^{c_4} \tag{22}$$

$$- 2 - a_4 - 2c_4 = 0$$

$$1 + a_4 + b_4 - c_4 = 0$$

$$1 + c_4 = 0$$

$$=> a_4 = 0, b_4 = 0, c_4 = -1$$

$$\pi_4 = \frac{F}{H_w} \tag{23}$$

$$\pi_5 = \theta(v_c)^{a_5}(f)^{b_5}(H_w)^{c_5} = ML^2T^{-2}\left(LT^{-1}\right)^{a_3}(L)^{b_3}\left(ML^{-1}T^{-2}\right)^{c_3} \tag{24}$$

$$- 2 - a_5 - 2c_5 = 0$$

$$2 + a_5 + b_5 - c_5 = 0$$

$$1 + c_5 = 0$$

$$=> a_5 = 0, b_5 = -3, c_5 = -1$$

$$\pi_5 = \frac{\theta}{f^3 H_w} \tag{25}$$

$$Z\frac{v_c}{f} = G\left(\frac{a}{f}, \frac{H_T}{H_w}, \frac{F}{H_w}, \frac{\theta}{f^3 H_w}\right) \tag{26}$$

$$T = \frac{1}{v_c} G\left(a, \frac{H_T f}{H_w}, \frac{F f}{H_w}, \frac{\theta}{f^2 H_w}\right) \tag{27}$$

Formula (27) shows the dominant effect of the cutting speed on all the dimension groups, which means that the tool life, cutting force, and cutting temperature are influenced by the cutting force. The relationship of the tool life with the depth of cut is linear. The tool life and tool hardness are directly related. The hardness of the workpiece material is the second parameter which plays a significant role in estimating the tool life in small-lot production. This function demonstrates the importance of considering the workpiece material hardness in small-lot production when one tool performs operations on the different workpieces. The effect of the cutting temperature on the tool life is non-linear with respect to the feed rate, which makes it more difficult to predict its role in estimating the tool life.

## 4. Discussion

This paper studies the parameters that affect tool life in the metal turning process. The direct and indirect effects of these parameters were extracted from 101 papers and books. Much literature has investigated the role of machining parameters in producing a product. Substantially, the cutting speed has the dominant effect on various parameters, such as the cutting force, cutting temperature, vibration, friction, and, finally, tool wear. The Taylor equation is the outcome of these studies, using cutting conditions to calculate the tool life. The impacts of the tool and workpiece material are considered in the constants of the equation. These constants are obtained from experience and studies in the literature. This approach is useful in mass production or large-lot-size production, using one tool for one specific workpiece type.

In small-lot production, the same tool may operate on different workpiece materials. Therefore, other parameters are taken into account when calculating the tool life. This article studied parameters directly and indirectly affecting tool wear and tool life in the metal turning process. The results of the literature review are demonstrated in a relationship matrix. The matrix enables the extraction of dominant parameters and could be useful for further studies. In small-lot production, the main changes occur in the workpiece material and geometry. When the workpiece changes, the cutting parameters change, which affects other parameters. This study concentrated on the effects of workpiece material properties on tool life, both directly and indirectly.

Estimating tool life in small-lot production affects production resource consumption, costs, and time. Tool failure during production causes long downtimes and maintenance needs and increases the cost and time of production. In small- and medium-sized enterprises, the production plan is often based on the experience of the machine operator. The operator must organize the parts with different geometries and materials. To save time and reduce downtime, the parts can be organized so as to use the same tool to manufacture consecutive parts. This reduces the amount of unproductive time, working time, and energy consumption. Knowing how different parameters affect tool life helps the operator to develop an efficient production plan for manufacturing small lots. Figure 13 illustrates an example of the small-lot production of three different types of workpieces (different geometric shapes are used as symbols of different workpieces) which can be manufactured with one tool.

The graph-based analysis illustrates the weights of the parameters affecting tool life. The cutting temperature, axial and radial forces, chip formation, chip thickness, cutting speed, feed rate, and rake angle are the most weighted parameters. In order to study tool life in small-lot production, it is important to consider which parameters are affected by the workpiece material and geometry. The cutting conditions are calculated according to the workpiece material properties and geometry. In the manufacturing of different products

with the same tool, the cutting force, cutting temperature, friction, chip formation, and chip thickness vary. Consequently, differences in these factors cause different wear behaviors on the tool inserts.

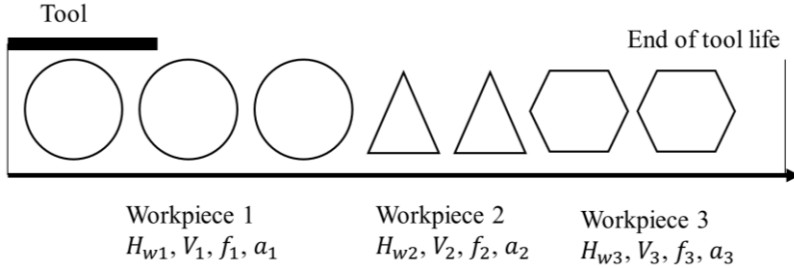

**Figure 13.** Manufacturing in small-lot production with a tool.

In this research, we used the outcomes of the literature review to extract important parameters in small-lot production so as to determine a formula that can be used to calculate tool life with changing parts in small-lot production. These parameters were used for dimensional analysis to derive a tool life formula for small-lot production. In small-lot production, the tool life depends on the cutting conditions and tool and workpiece hardness. When using a tool for different workpieces, the cutting conditions may be different, which changes the cutting force, vibration, friction, and cutting temperature. Normally, a tool is selected for the production according to the range of cutting conditions and workpiece properties. The dominant wear occurs in this range, which means that the dominant tool wear, in small-lot production, does not change according to the selected tool.

Small-lot production is increasing in Nordic countries and Europe. It has resulted in demands for the maintenance of machine equipment, customized products, and spare part manufacturing. These sectors benefit from the calculation and monitoring of tool life to optimize production planning.

## 5. Conclusions

In this study, we employed dimensional analysis to formulate the tool life in the turning process of metal cutting for small-lot production by considering the effects of the most significant parameters. Dimensional analysis of the tool life involves the cutting speed, feed rate, depth of cut, workpiece hardness, tool hardness, cutting force, and cutting temperature. The final tool life function shows that the cutting speed, workpiece material hardness, and feed rate are the most important factors for estimating the tool life.

The achieved function involves the relationships between the used parameters and tool life. The effects of the cutting force and cutting temperature are non-linear. In agreement with the established theory, increasing the tool hardness increases the tool life, while a higher workpiece hardness reduces the tool life. Coefficients need to be defined experimentally in future work. By understanding the relationships between the factors that impact tool wear and tool life, manufacturers could develop the production plans that use the optimized tool life in small-lot production, and the results of this study could enable different approaches to the estimation of tool life, including artificial intelligence development, big data analysis, and digital twins.

These parameters were extracted from a literature review of 101 references, including books, conference papers, and scientific articles from 2000 to 2022. The outcomes of the literature review are a relationship matrix and a weighted graph based on this matrix. The relationship matrix demonstrates 29 parameters affecting 23 factors which directly or indirectly influence tool life. The graph also illustrates the weights of these factors affecting the tool life. A cause–effect diagram was developed from the literature review to study tool life in small-lot production.

**Author Contributions:** Conceptualization, M.L. and S.M.B.; methodology, S.M.B. and J.R.; software, S.M.B.; validation, M.L., J.R. and J.V.; formal analysis, S.M.B.; investigation, S.M.B.; writing—original draft preparation, S.M.B.; writing—review and editing, J.R. and M.L.; visualization, J.R. and M.L.; supervision, J.R., J.V. and M.L.; project administration, M.L. and J.V. All authors have read and agreed to the published version of the manuscript.

**Funding:** This research received no external funding.

**Data Availability Statement:** Not applicable.

**Conflicts of Interest:** The authors declare no conflict of interest.

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
