# Peer review of "An Investigation of Factors Influencing Tool Life in the Metal Cutting Turning Process by Dimensional Analysis†"

_machines, doi:10.3390/machines11030393_

Round 1

Reviewer 1 Report

The article focuses on Investigation into factors influencing the tool life in the metal cutting turning process based on the literature review. The intention of the work is good although to enhance the quality of the manuscript, I have some comments and suggestions.

1.       Cutting environment such as dry, MQL, and wet and also residual stress are other influential factors on the tool wear and life. The authors can also consider these factors on the tool life.

2.       The references should be more discussed instead of being cited successively.

3.       To increase the quality of the introduction section, you need to add some recently published papers on the influence of turning parameters on tool wear and life.

4.       A deeper discussion of the results obtained is necessary.

In conclusion, If the authors apply the corrections to the manuscript within due time, the manuscript will be suitable for publication in "Machines". 

Updated review report:    

The article focuses on Investigation into factors influencing the tool life in the metal cutting turning process based on the literature review. The intention of the work is good although to enhance the quality of the manuscript, I have some comments, suggestions, and questions. 

1.  What is the added value of the present model compared to those that exist in the literature?

2.  To increase the quality of the introduction section, you need to add some recently published papers on the influence of turning parameters on the tool wear and life. 

3.  The authors need to highlight their contribution and novelty since it is not clearly mentioned in the manuscript. You need to add a paragraph at the end of the Section “Introduction” and mention your contribution with respect to other published papers. 

4.  The references should be more discussed instead of being cited successively.

5.  Cutting environment such as dry, MQL, and wet and also residual stresses are another influential factors on the tool wear and life. The authors can also consider these factors on the tool life. 

6.  Are there any references for Figures 6 and 7? Please explain why these two figures are important to be mentioned in the manuscript. 

7.  Sections “Discussion” and “Conclusions” should be improved based on the obtained results. 

In conclusion, If the authors apply the corrections to the manuscript within due time, the manuscript will be suitable for publication in "Machines".

Author Response

Thank you for giving us the opportunity to submit a revised draft of the manuscript “An Investigation into Factors Influencing the Tool Life in the Metal Cutting Turning process by Dimensional Analysis” for publication in the Machine Journal. We appreciate the time and effort that you and the reviewers dedicated to providing feedback on our manuscript and are grateful for the insightful comments on and valuable improvements to our paper. We have incorporated most of the suggestions made by the reviewers. Those changes are highlighted within the manuscript. Please see below for a point-by-point response to the reviewers’ comments and concerns. All page numbers refer to the revised manuscript file with tracked changes.

Reviewer 2 Report

General remarks:

1. First of all, I would like to congratulate the authors for the quality of their work. The work is interesting, although there are several similar works that study aspects close to this work. The originality lies in focusing systematically on the function of the tool life for the small lot production. The objectives have been covered and the paper reads well and is adequately structured. In the opinion of the reviewer, the paper presented is mostly well written and interesting, but it must overcome some important drawbacks.  

2. The main idea behind the paper is interesting: studying the influencing factors on the tool life in the metal cutting turning process using dimensional analysis. Dimensional analysis is executed based on the cause-and-effect diagram to calculate tool life.

3. The title and the intentions declared in the abstract correspond to the contents of the paper. The paper contains an abstract and introduction which is in fact a critical review of the state of the art. The authors have important contributions in the field of turning process.

Specific remarks:

4. How and why can your research work contribute to the strengthening of science, engineering, and technological development of a particular society or scientific community? Which is the environmental, social, and economic impact of your research work or study make?

5. I found many similarities in the present paper with a conference paper of the same authors, but I understood that, in fact, the paper was selected for publishing from this conference.

6. Where is the novelty in this study with respect to other papers? What is their contribution, with this research work, to the academic-scientific community of sciences and engineering?

7. At the end of the Introduction chapter the authors must mention which is the novelty of the paper with respect to the papers presented in the state of the art. Which are the strong points of the present paper? What does the paper brings new related to other papers?

8. Page 3, line 105, please correct this phrase “Section concludes the results if the study”.

9. Page 3, line 113, I consider that the description regarding the Gephi application ([https://gephi.org/]) must be put as a reference, not directly in the text of the paper.

10. Page 4, line 157. Can the authors explain why they chose an exclusive criterion like EC2 (The paper should not be about AI methods to estimate tool life).

11. Page 4, Figure 1. Can the authors explain why they exclude keywords such as ”surface properties” and ”morphology”? What was the cause of these exclusions because in my opinion these keywords could affect the results of the analysis?

12. I consider that Figures 2 and 3 are redundant. I recommend keeping only one of them.

13. Page 6, Figure 7, please review the caption of the figure Number of studies the important parameters affecting tool life in the database”.

14. Page 7, lines 215-216. Please capitalize the first letter of the sentence: ”the aim of this study is to show there is a relationship between parameters, not how this relationship is in different special cases”

15. The discussion chapter is a little bit superficial. I consider that the authors can improve this chapter.

16. Finally, the conclusions section is just a summary of the main results that without a previous discussion of results is just a summary of the results.

Author Response

(The authors gave the same response as above.)

Reviewer 3 Report

Dear authors,

see the comments in attachment.

BR 

Author Response

(The authors gave the same response as above.)

Round 2

Reviewer 3 Report

Dear authors,

you have made almost all correction, except to correct the equation nr. 5. (Vs should be vs).

Best regards

Author Response

Hello

Thank you for pointing it out. It is corrected in the revised manuscript. 

Best regards